# Comparison of Clinical Characteristics and Outcomes between Idiopathic and Secondary Pleuroparenchymal Fibroelastosis

**DOI:** 10.3390/jcm10040846

**Published:** 2021-02-18

**Authors:** Tsuneyuki Oda, Akimasa Sekine, Erina Tabata, Tae Iwasawa, Tamiko Takemura, Takashi Ogura

**Affiliations:** 1Kanagawa Cardiovascular and Respiratory Center, Department of Respiratory Medicine, Yokohama 236-8651, Japan; Akimasa.Sekine@gmail.com (A.S.); e-tabata@kanagawa-junko.jp (E.T.); ogura@kanagawa-junko.jp (T.O.); 2Kanagawa Cardiovascular and Respiratory Center, Department of Radiology, Yokohama 236-8651, Japan; iwasawa@kanagawa-junko.jp (T.I.); tamikobyori@gmail.com (T.T.)

**Keywords:** pleuroparenchymal fibroelastosis, UIP, hypersensitivity pneumonitis, interstitial lung diseases

## Abstract

Background: Pleuroparenchymal fibroelastosis (PPFE) is a unique clinical, radiologic, and histopathologic entity for which several potential etiologies have been reported recently. However, there has been no comprehensive study of secondary PPFE. Objective: Assessment of the clinical characteristics, outcomes, and prognostic factors of secondary and idiopathic PPFE. Methods: We retrospectively reviewed the medical records of consecutive PPFE patients between January 1999 and December 2018. We identified 132 idiopathic PPFE patients and 32 secondary PPFE patients. Results: The incidence of interstitial lung disease (ILD) pattern different from the usual interstitial pneumonia (UIP) pattern in the lower lobes was higher in secondary PPFE patients (38.5%) than in idiopathic PPFE patients (61.5%, *p* = 0.02). The idiopathic and secondary PPFE groups did not differ significantly in terms of laboratory data, respiratory complications, and survival (median: 5.0 years vs. 4.1 years, *p* = 0.95). The presence of UIP pattern was independently associated with increased mortality in multivariate analyses in idiopathic PPFE patients, but not in secondary PPFE patients. Conclusions: The frequency and prognostic impact of UIP-pattern ILD differed between idiopathic and secondary PPFE patients. Lung transplantation should be considered in secondary PPFE patients with low diffusing capacity of the lungs for carbon monoxide (DLCO) regardless of lower-lobe ILD pattern.

## 1. Introduction

Pleuroparenchymal fibroelastosis (PPFE) was first reported as a unique pleuroparenchymal lung disease in 2004 [1]. Idiopathic PPFE has been reported to account for approximately 6–8% of all interstitial lung diseases; however, the true prevalence of PPFE is not known. Several recent studies have explored and documented the possible etiologies of secondary PPFE, including radiation [1], bone marrow or stem cell transplantation [2,3], lung transplantation [2,4], connective tissue diseases [5,6,7], asbestosis [8], pneumoconiosis [8,9], metal lung [10,11], recurrent infections (e.g., pulmonary aspergillus [12] or Mycobacterium avium intracellular complex [13]), chemotherapeutic agents [14,15], and chronic hypersensitivity pneumonitis (HP) [12,16,17]. The diagnosis of secondary PPFE is challenging because of the contraindications for surgical lung biopsy. Surgical lung biopsy is associated with risks such as a potentially prolonged period of post-operative iatrogenic pneumothorax [18,19]. There has been no comprehensive study of secondary PPFE. Thus, the clinical differences between idiopathic and secondary PPFE remain unknown.

Prognostic factors for PPFE reported in previous studies include the Krebs von den Lungen 6 (KL-6) [20] levels, body mass index, predicted forced vital capacity (FVC) % [21], and the gender-age-physiology index score [21]. However, the presence of the usual interstitial pneumonia (UIP) pattern as a predictor remains controversial. Several studies have shown that the UIP pattern in the lower lobes was a poor prognostic factor in idiopathic PPFE [20,22,23], whereas others determined it to be insignificant [16,24,25].

In this retrospective study, we aimed to determine and elucidate the clinical characteristics, outcomes, and prognostic factors of secondary PPFE compared with idiopathic PPFE in a large population of patients.

## 2. Materials and Methods

### 2.1. Patient Selection and Data Collection

From the Kanagawa Cardiovascular and Respiratory Center research database, we retrospectively identified 68 PPFE patients who underwent surgical lung biopsy or autopsy between January 1999 and December 2018 and 96 PPFE patients who did not undergo surgical lung biopsy or autopsy between January 2012 and December 2018.

Patients who met the radiologic and the pathologic criteria (if biopsy specimens were available) for the diagnosis of PPFE, as described below, were subjected to an evaluation and multidisciplinary discussion involving radiologists, pathologists, and pulmonologists. Patients were divided into two subgroups: Idiopathic PPFE group and secondary PPFE group with a known cause or association (Figure 1).

Individual differentiated connective tissue diseases, such as Sjögren’s syndrome, systemic sclerosis, and rheumatoid arthritis, were diagnosed using standard criteria [26,27,28]. HP was diagnosed based on the official criteria of the American Thoracic Society, the Japanese Respiratory Society, and the Latin American Thoracic Association clinical practice guideline [29]. Clinical and laboratory data, pulmonary function test (PFT) data, and bronchoalveolar lavage (BAL) fluid findings on initial examination were collected retrospectively from the medical records. Data regarding the clinical course of each patient after diagnosis, including complications and prognosis, were also recorded.

The protocol for this study was approved by the institutional review board at the Kanagawa Cardiovascular Respiratory Center (approval number KCRC-19-0006). The Ethical Committee waived the requirement for informed consent due to the retrospective nature of the study. The study cohort overlapped partially with the cohort described in a previous publication [20,30].

### 2.2. High-Resolution Computed Tomography Evaluation

A pulmonary radiologist (T.I.) and a pulmonologist (T.O.) reviewed the chest computed tomography (CT) data collected from all patients on initial examination. The reviewers were blinded to the patients’ clinical background and histopathologic data. The presence of a UIP pattern in PPFE patients was assessed on high-resolution CT according to the guidelines of the American Thoracic Society, the European Respiratory Society, the Japanese Respiratory Society, and the Latin American Thoracic Association [31]. Any disagreement in the final diagnoses was resolved by consensus.

According to Lee et al. [32], the radiological criteria for PPFE were defined as follows: (1) Bilateral subpleural dense fibrosis with or without pleural thickening in the upper lobes; and (2) confirmation of disease progression (defined as an increase in the upper lobe consolidation with or without pleural thickening and/or a decrease in upper lobe volume on serial radiological assessment).

### 2.3. Histopathologic Criteria for PPFE

The histopathologic data were reviewed independently by a pathologist who was blinded to the clinical findings. In this study, the following histopathological diagnostic criterion for PPFE was applied [1,12,33]: Upper zone pleural fibrosis with subjacent intra-alveolar fibrosis accompanied by alveolar septal elastosis. The histopathologic diagnoses of UIP [31] pattern and non-specific interstitial pneumonia (NSIP) [34] pattern was determined according to the guidelines.

### 2.4. Statistical Analyses

The baseline characteristics of patients are presented as medians (interquartile ranges) for continuous variables and as numbers for categorical variables. These baseline variables were compared between groups using Fisher’s exact test (categorical variables) and the Mann–Whitney U test (continuous variables). Survival analysis was performed using the Kaplan–Meier method, and the estimates were compared between groups using the log-rank test. Survival time was calculated as the time from the initial patient consultation to death or last contact. The survival outcomes of the patients were followed until 30 May 2020.

Cox proportional hazards model analysis was used to detect independent predictors of survival. The following factors were included in our univariate analysis: Age, sex, body mass index, PFT data, KL-6, BAL fluid findings, and lower-lobe interstitial lung disease (ILD) patterns. Subsequently, variables with a *p* value < 0.05 in the univariate analysis were included in the multivariate analysis. For all statistical tests, a two-tailed *p* value of < 0.05 was considered statistically significant. SPSS software, version 24 (IBM Corp., Armonk, NY, USA), was used to perform the statistical analyses.

## 3. Results

### 3.1. Clinical Characteristics, PFTs, and BAL Fluid Findings

A total of 164 patients with idiopathic and secondary PPFE were included in this study. The median age of the 81 men and 83 women was 67 years (interquartile range = 16 years) on initial examination. Sixty-six (40.2%) patients had a history of smoking, with a median of 16.9 pack-years. The median follow-up period for the overall population was 2.1 years (interquartile range = 2.8 years).

Out of the 164 patients, 32 patients had secondary PPFE. Of these, chronic HP was the most associated disease (*n* = 10, Figure 2 and Figure 3), followed by Sjögren’s syndrome (*n* = 6), pneumoconiosis (*n* = 4), metal lung (*n* = 2), non-tuberculosis mycobacterial infection (*n* = 2), rheumatoid arthritis (*n* = 2), and chemotherapeutic agents (*n* = 2); data are presented in Table 1.

Baseline characteristics, lower-lobe ILD pattern, and respiratory complications of the patients with idiopathic and secondary PPFE are presented in Table 2. Patients with secondary PPFE were younger than those with idiopathic PPFE (68.5 years vs. 59.5 years, *p* < 0.001). In the BAL fluid findings, the secondary PPFE group had a significantly lower percentage of macrophages (*p* = 0.003) and a higher percentage of lymphocytes (*p* = 0.001) than the idiopathic PPFE group. The incidence of ILD pattern different from the usual interstitial pneumonia (UIP) pattern in the lower lobes was higher in secondary PPFE patients than in idiopathic PPFE patients (*p* = 0.02). No significant differences were observed between the two groups in terms of symptoms on initial examination, laboratory data, PFT, and respiratory complications during the follow-up period. No lung cancer developed in either group. The overall number of deaths was 61 patients with a mortality rate of 37.2% during follow-up periods. Idiopathic PPFE patients had a mortality rate of 34.1%, and secondary PPFE patients had a mortality rate of 50%.

### 3.2. Predictor of Mortality for Idiopathic and Secondary PPFE

The Cox proportional hazards model was used to identify predictors of mortality in patients with idiopathic and secondary PPFE. A univariate analysis of idiopathic PPFE patients identified that age, FVC, diffusing capacity of the lungs for carbon monoxide (DLCO), residual volume/total lung capacity (RV/TLC), KL-6, and the presence of UIP pattern in the lower lobes were significantly associated with poor prognosis. In the multivariate analysis, FVC, DLCO, and the presence of UIP pattern in the lower lobes were identified as independent predictors of mortality in idiopathic PPFE patients (Table 3, Figure 4A). In secondary PPFE patients, univariate and multivariate analysis identified that DLCO was significantly associated with poor prognosis; however, FVC and the presence of UIP pattern in the lower lobes did not predict mortality (Table 4, Figure 4B). A univariate analysis of the patients with the entire cohort identified age, FVC, DLCO, Krebs von den Lungen 6, and the UIP pattern in the lower lobes significantly associated with a poor prognosis. A diagnosis of secondary PPFE did not predict mortality (Table 5).

### 3.3. Survival Analysis of Patients with Idiopathic and Secondary PPFE

A comparison of the survival analysis of idiopathic and secondary PPFE patients is shown in Figure 5. Survival curves for the two groups were similar (log-rank *p* = 0.95). The median survival times of idiopathic and secondary PPFE were 5.0 years and 4.1 years, respectively.

## 4. Discussion

The aim of this study was to assess the clinical characteristics, outcomes, and prognostic factors of idiopathic and secondary PPFE in a large population of patients. The main findings of this study suggest two important clinical issues. First, clinical characteristics and outcomes in secondary PPFE patients were similar to those in idiopathic PPFE patients, except for age, BAL fluid findings, and lower-lobe ILD pattern. Second, the presence of UIP pattern was a significant poor predictor in the idiopathic PPFE group, but not in the secondary PPFE group.

As mentioned above, clinical characteristics, physiology, and respiratory complications in secondary PPFE patients were similar to those in idiopathic PPFE patients. Since secondary PPFE is a form of underlying cause, it can be assumed that the secondary PPFE group would be younger and would have a wider variety of ILD patterns, such as NSIP pattern and unclassifiable interstitial pneumonia pattern in the lower lobes, than the idiopathic PPFE group. Survival curves for the idiopathic and secondary PPFE groups were similar. These findings suggest that idiopathic PPFE as well as secondary PPFE have a poor prognosis and that lung transplantation should be considered in young PPFE patients with advanced stages, even in secondary PPFE patients.

In patients with idiopathic PPFE, the presence of UIP pattern in the lower lobes was independently associated with increased mortality, whereas the prognostic factor for secondary PPFE was not the presence of UIP pattern, but DLCO. Previous studies have described a shorter survival time in PPFE patients with UIP pattern than in idiopathic pulmonary fibrosis patients [21,30,32]. The progression of disease in both the upper and lower lobes may lead to poorer prognosis in PPFE patients with UIP pattern. Although it remains unclear why secondary PPFE patients with UIP pattern do not present with poor prognoses, chest physicians should be aware that the effect of UIP-pattern ILD on prognosis differs between idiopathic and secondary PPFE.

The presence of the UIP pattern as a predictor remains controversial. Previous studies on prognostic factors of PPFE may have included a mixture of idiopathic and secondary PPFE in the population. Therefore, we speculate that the UIP pattern of the lower lobe be a predictor depending on the different studies. It is difficult to identify the cause of secondary PPFE. In this study, we tried to identify the cause of secondary PPFE as much as possible by blood tests and pathology. Furthermore, in the previous studies on prognostic factors of PPFE, there were a few cases where the diagnosis of UIP pattern was confirmed by histopathology. In the present study, many patients were diagnosed by histopathology, which may be another reason for the different results of studies.

Conditions that primarily affect the pulmonary vasculature, such as pulmonary hypertension and pulmonary embolism, decrease DLCO. In this study, there remain unanswered questions concerning the etiology of reduced DLCO in PPFE patients. Further study is required to examine the etiology of reduced DLCO with echocardiography and contrast-enhanced computed tomography to screen for the presence of pulmonary hypertension and pulmonary embolism.

Chronic HP was relatively common in the secondary PPFE group (10/32 patients). Despite the apparent possible link between the pathogenesis of PPFE and HP, we were unable to evaluate specific inhaled antigens or allergens in this study. Khiroya et al. speculated that PPFE might represent a progressive fibrosing immune-mediated response to an identified or unidentified inhaled antigen or allergen. Further studies are required to examine the association between the pathogenesis of PPFE and HP.

This study had several limitations, including the single center setting and retrospective design. Our results may not be representative of the entire population with secondary PPFE because we did not encounter the secondary PPFE with chronic pulmonary aspergillus, and PPFE relapse after lung transplantation, and so on. The numbers of patients with secondary PPFE is so small that strong conclusions on mortality risk cannot be drawn. The small number of patients in each secondary PPFE subgroup significantly limited our exploratory analyses.

In conclusions, the clinical features and outcomes of secondary PPFE patients were similar to those of idiopathic PPFE patients. The presence of UIP pattern in the lower lobes was a poor prognostic factor in idiopathic PPFE, but not in secondary PPFE. Secondary PPFE had a poor prognosis, and lung transplantation should be considered in secondary PPFE patients with low DLCO regardless of lower-lobe ILD pattern.

## Figures and Tables

**Figure 1 jcm-10-00846-f001:**
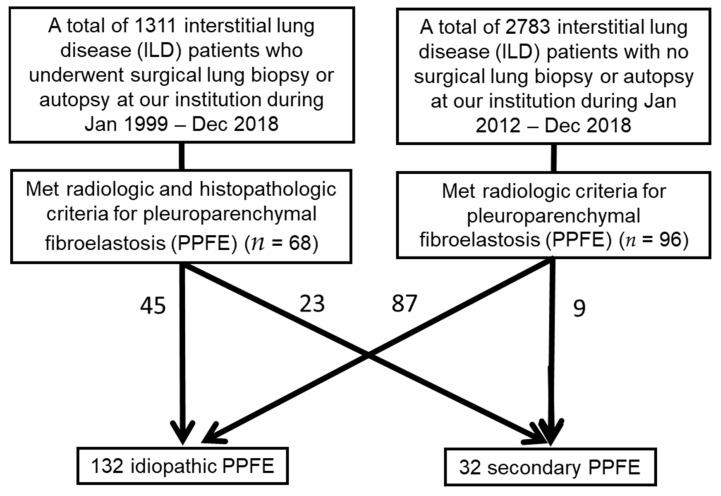
Flow sheet. ILD, interstitial lung disease; PPFE, pleuroparenchymal fibroelastosis.

**Figure 2 jcm-10-00846-f002:**
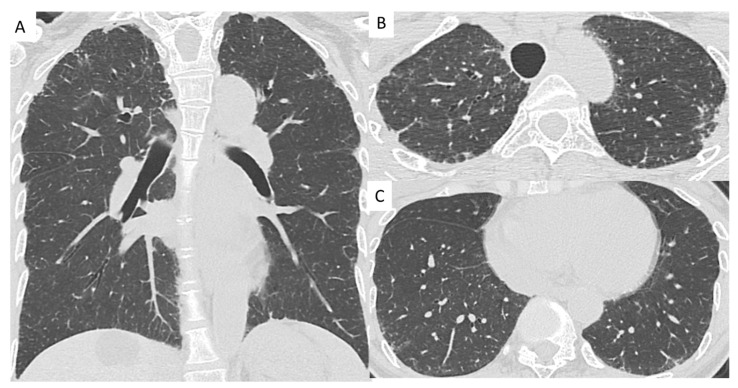
Chest computed tomography images from a patient with pleuroparenchymal fibroelastosis (PPFE) with hypersensitivity pneumonitis. (**A**) Chest computed tomography shows bilateral dense subpleural consolidation and a loss of upper lobe volume with hilar elevation in the coronal plane. (**B**) There are subpleural consolidation in the upper lobes. (**C**) Chest computed tomography shows ground-glass opacity and reticulation in the subpleural predominance of the lower lobes.

**Figure 3 jcm-10-00846-f003:**
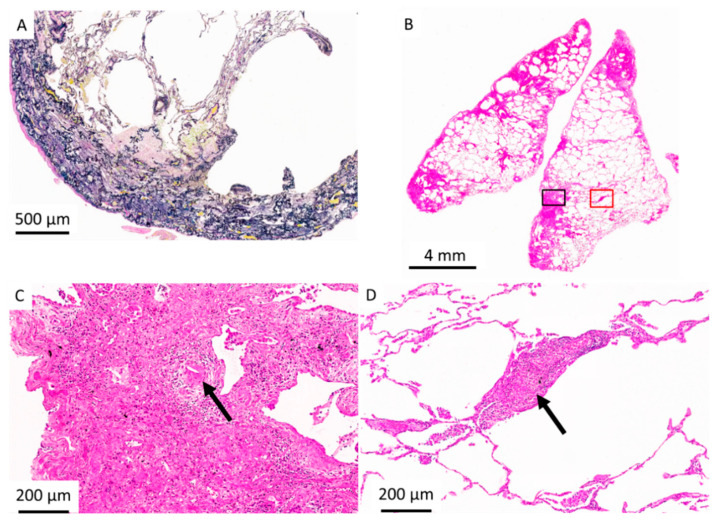
Surgical lung biopsy specimen from two different sites in a patient with PPFE with hypersensitivity pneumonitis. (**A**) High-magnification photomicrograph of a left-upper-lobe biopsy specimen shows subpleural fibroelastosis (Elastic van Gieson staining). (**B**) Panoramic view of a left-lower-lobe biopsy specimen shows subpleural and paraseptal predominant fibrosis (Hematoxylin and eosin staining). (**C**) High-magnification photomicrograph of subpleural fibrosis (black square in Figure 2B) illustrates fibroblastic foci, mononuclear inflammatory cells, and a giant cell with cholesterol cleft (arrow) (hematoxylin and eosin staining). (**D**) High-magnification photomicrograph of red square in Figure 2B illustrates poorly formed non-necrotizing granuloma (arrow) and lymphocytic infiltration around a respiratory bronchiole (hematoxylin and eosin staining).

**Figure 4 jcm-10-00846-f004:**
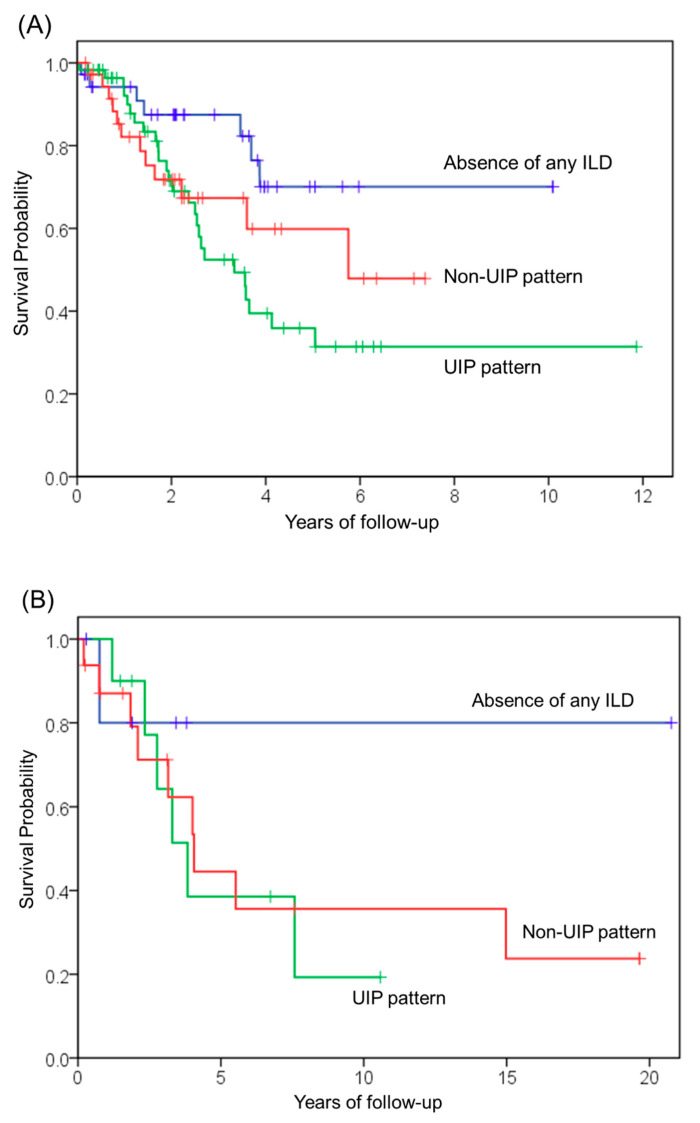
Subgroup survival analyses of patients with PPFE. Survival durations were estimated and compared using the Kaplan–Meier method and log-rank test, respectively. (**A**) Idiopathic PPFE patients were divided into three groups based on the following: (1) The presence of UIP pattern; (2) the presence of non-UIP pattern; (3) the absence of any ILD pattern in the lower lobes. Absence of any ILD versus non-UIP pattern (*p* = 0.16); absence of any ILD versus UIP pattern (*p* = 0.01); non-UIP pattern versus UIP pattern (*p* = 0.42). (**B**) Secondary PPFE patients were divided into three groups based on the following: (1) The presence of UIP pattern; (2) the presence of non-UIP pattern; (3) the absence of ILD pattern in the lower lobes. There were no statistical differences between the groups. Absence of any ILD versus non-UIP pattern (*p* = 0.33); absence of any ILD versus UIP pattern (*p* = 0.34); non-UIP pattern versus UIP pattern (*p* = 0.78).

**Figure 5 jcm-10-00846-f005:**
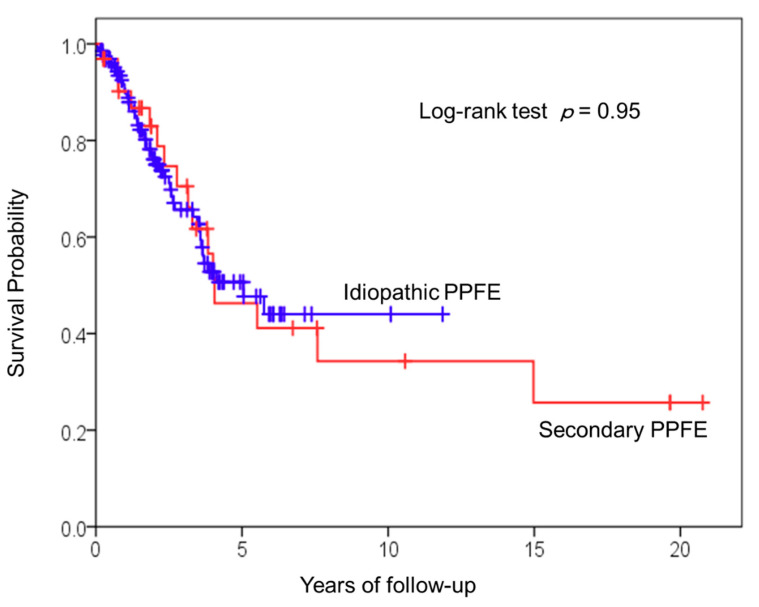
Survival analysis of idiopathic versus secondary PPFE. Kaplan-Meier survival curves show the number of idiopathic PPFE patients versus secondary PPFE patients. Survival curves for the two groups were similar (log-rank test, *p* = 0.95).

**Table 1 jcm-10-00846-t001:** Data presenting proven causes of secondary PPFE.

Cause of Secondary PPFE	*n* (%)
Chronic hypersensitivity pneumonitis	10 (31.3)
Sjögren’s syndrome	6 (18.8)
Pneumoconiosis	4 (12.5)
Rheumatoid arthritis	2 (6.3)
Metal lung	2 (6.3)
Mycobacterium infection	2 (6.3)
Chemotherapeutic agents	2 (6.3)
Systemic sclerosis	1 (3.1)
Crohn’s disease	1 (3.1)
Bone marrow transplantation	1 (3.1)
Radiation	1 (3.1)

Data are presented as numbers (%). PPFE; pleuroparenchymal fibroelastosis. Percentages are rounded off to the first decimal place.

**Table 2 jcm-10-00846-t002:** Comparison of the baseline characteristics between idiopathic and secondary PPFE.

Parameters	Idiopathic PPFE (*n* = 132)	Secondary PPFE			
		All (*n* = 32)	Chronic HP (*n* = 10)	Sjögren’s Syndrome (*n* = 6)	Others (*n* = 16)
**Demographics**					
Age, years	68.5 (14)	59.5 (13) *	63 (14)	60 (13)	55 (17)
Sex, male *n* (%)	68 (51.5)	13 (40.6)	4 (40)	2 (33.3)	7 (43.7)
Smoker *n* (%)	57 (43.2)	9 (28.1)	4 (40)	1 (20)	4 (25)
Body mass index, kg/m^2^	19.8 (4.1)	19.7 (2.6)	20.3 (3.5)	17.2 (4.5)	18.2 (3.7)
Follow-up periods, years	2.1 (2.7)	3.1 (4.9)	4.8 (6.8)	3.6 (6.9)	2.2 (2.9)
**Diagnosis**					
Biopsy-proven/ No biopsy-proven, *n*	45/87	23/9 *	10/0	4/2	9/7
**Symptoms on initial examination**					
No symptoms, *n* (%)	29 (22.0)	9 (28.1)	3 (30)	3 (50)	3 (18.8)
Cough, *n* (%)	55 (41.7)	14 (43.8)	4 (40)	1 (16.7)	9 (56.3)
Dyspnea on exertion, *n* (%)	76 (57.6)	13 (40.6)	3 (30)	2 (33.3)	8 (50)
**Pulmonary function test data**					
FVC, % predicted	73.0 (31.6)	70.0 (21.7)	73.4 (14.0)	65.9 (27.9)	52.5 (30.9)
DLCO, % predicted	89.4 (33.1)	78.8 (59.1)	97.6 (63.0)	88.3 (51.3)	60.9 (31.8)
RV/TLC, %	45.2 (16.9)	44.9 (11.3)	39.5 (16.6)	44.3 (14.5)	47.5 (12.0)
**Laboratory data**					
KL-6, IU/mL	638.5 (447)	557.5 (290)	544 (299)	529.5 (374)	593.5 (552)
BAL, Macrophages, %	83.8 (16.5)	62.9 (22.4) ^†^	59.5 (13.8)	80.0 (*n* = 3)	69.8 (21.3) (*n* = 8)
BAL, Lymphocytes, %	11.3 (11.7)	23.9 (17.9) ^‡^	31.3 (19.5)	17.2 (*n* = 3)	19.4 (16.0) (*n* = 8)
**Lower-lobe ILD pattern**					
UIP pattern, *n* (%)	59 (44.7)	10 (31.3) ^§^	5 (50)	3 (50)	2 (12.5)
NSIP patten, *n* (%)	5 (3.8)	4 (12.5)	1 (10)	2 (33.3)	1 (6.3)
Absence of any ILD in the lower lobes, *n* (%)	36 (27.3)	6 (18.8)	0	0	6 (37.5)
**Respiratory complications**					
Pneumothorax or pneumomediastinum, *n* (%)	63 (47.7)	16 (50)	5 (50)	5 (50)	6 (37.5)
Acute exacerbation, *n* (%)	5 (3.8)	2 (6.3)	1 (10)	0	1 (6.3)
Dead, *n* (%)	45 (34.1)	16 (50)	3 (30)	5 (83.3)	8 (50)

Data are presented as median (interquartile range) or numbers (%). HP, hypersensitivity pneumonitis; FVC, forced vital capacity; DLCO, diffusing capacity of the lung for carbon monoxide; RV, residual volume; TLC, total lung capacity; KL-6, Krebs von den Lungen 6; BAL, bronchoalveolar lavage; UIP, usual interstitial pneumonia; ILD, interstitial lung disease. * *p* < 0.001 for idiopathic PPFE versus secondary PPFE; ^†^
*p* = 0.003 for idiopathic PPFE versus secondary PPFE; ^‡^
*p* = 0.001 for idiopathic PPFE versus secondary PPFE; ^§^
*p* = 0.02 for idiopathic PPFE versus secondary PPFE.

**Table 3 jcm-10-00846-t003:** Cox proportional hazards model assessment of potential predictors of mortality in patients with idiopathic PPFE.

Parameters	Univariate Analysis		Multivariate Analysis	
	Hazard Ratio (95% CI)	*p* Value	Hazard Ratio (95% CI)	*p* Value
Age	1.04 (1.01–1.07)	0.02	1.04 (0.99–1.08)	0.09
Sex, male	1.58 (0.87–2.87)	0.14		
Body mass index	0.93 (0.83–1.04)	0.21		
FVC, % predicted	0.94 (0.93–0.96)	<0.01	0.93 (0.90–0.96)	<0.01
DLCO, % predicted	0.96 (0.94–0.97)	<0.01	0.98 (0.960–0.999)	0.04
RV/TLC	1.04 (1.00–1.07)	0.03	0.93 (0.94–1.03)	0.51
KL-6	1.001 (1.000–1.001)	<0.01	1.001 (1.000–1.002)	0.18
Lymphocyte % in the BAL fluid	0.99 (0.94–1.03)	0.50		
Presence of UIP pattern in the lower lobes	1.86 (1.03–3.37)	0.04	7.56 (3.39–16.9)	<0.01
Absence of any ILD in the lower lobes	0.40 (0.18–0.89)	0.03	0.93 (0.27–3.22)	0.93

CI, confidence interval; FVC, forced vital capacity; DLCO, diffusing capacity of the lung for carbon monoxide; RV, residual volume; TLC, total lung capacity; KL-6, Krebs von den Lungen 6; BAL, bronchoalveolar lavage; UIP, usual interstitial pneumonia; ILD, interstitial lung disease.

**Table 4 jcm-10-00846-t004:** Cox proportional hazards model assessment of potential predictors of mortality in patients with secondary PPFE.

Parameters	Univariate Analysis		Multivariate Analysis	
	Hazard Ratio (95% CI)	*p* Value	Hazard Ratio (95% CI)	*p* Value
Age	1.02 (0.97–1.08)	0.41		
Sex, male	1.34 (0.48–3.76)	0.59		
Body mass index	0.92 (0.75–1.14)	0.44		
FVC, % predicted	0.98 (0.95–1.01)	0.28		
DLCO, % predicted	0.95 (0.92–0.98)	<0.01	0.95 (0.92–0.99)	<0.01
RV/TLC	0.98 (0.93–1.04)	0.57		
KL-6	1.002 (1.000–1.003)	0.01	1.000 (0.999–1.002)	0.56
Lymphocyte % in the BAL fluid	0.95 (0.89–1.01)	0.09		
Presence of UIP pattern in the lower lobes	1.33 (0.47–3.74)	0.59		
Absence of any ILD in the lower lobes	0.35 (0.05–2.68)	0.31		

See Table 3 legend for expansion of abbreviations.

**Table 5 jcm-10-00846-t005:** Cox proportional hazards model assessment of potential predictors of mortality in PPFE patients with the entire cohort.

Parameters	Univariate Analysis		Multivariate Analysis	
	Hazard Ratio (95% CI)	*p* Value	Hazard Ratio (95% CI)	*p* Value
Age	1.03 (1.01–1.06)	0.02	1.04 (1.01–1.07)	0.01
Sex, male	1.46 (0.88–2.43)	0.14		
Body mass index	0.93 (0.84–1.03)	0.93		
FVC, % predicted	0.95 (0.94–0.97)	<0.01	0.96 (0.94–0.98)	<0.01
DLCO, % predicted	0.96 (0.95–0.97)	<0.01	0.97 (0.95–0.99)	<0.01
RV/TLC	1.03 (1.00–1.05)	0.08		
KL-6	1.001 (1.000–1.001)	<0.01	1.001 (1.000–1.002)	0.047
Lymphocyte % in the BAL fluid	0.98 (0.95–1.01)	0.23		
Presence of UIP pattern in the lower lobes	1.70 (1.03–2.83)	0.04	7.08 (3.27–15.3)	<0.01
Absence of any ILD in the lower lobes	0.39 (0.19–0.82)	0.01	1.72 (0.61–4.84)	0.30
Secondary PPFE	1.02 (0.56–1.84)	0.95		

See Table 3 legend for expansion of abbreviations.

## Data Availability

All data generated or analyzed during this study are included in this published article.

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
