# Peer review of "Comparison of Clinical Characteristics and Outcomes between Idiopathic and Secondary Pleuroparenchymal Fibroelastosis"

_jcm, 2021, doi:10.3390/jcm10040846_

Round 1
Reviewer 1 Report
Dear Editors,
Dear Athors,
I found the manuscript entitled "Comparison of clinical characteristics and outcomes between idiopathic and secondary pleuroparenchymal fibroelastosis" very interesting, thoughtful, well planned and presented. Although the authors discuss in the study limitations that the study was single center and retrospective, in my opinion presentation the group of 164 patients with very rare disease deserves congratulations.
I have only small comments and suggestions:
-Page 3, line 109: please delete "3.Results" at the very end of the sentence
-Page 4, Table 1: I suggest to check the numbers expressed as percentage, according to my calculations the sum is 100.2 at the moment.
-Page 5, Table 2 - what is the reason for different presentation of BAL results in Sjogrens group in parentheses? Is it beacuse the results are for 3 patients only?
-Page 6, Table 3 - please check data in the table. In my opinion, in the last column, two penultimate p-values have been shifted downwards
-What's the reason for different understanding of UIP as a prognostic factor among different studes? Does it depend on methodological differences or type of PPFE (idiopathic versus secondary one) -maybe it's worth to add a comment in the discussion?
-It could be also interesting if the authors could compare some physical findings (e.g. acultatory abnormalities, presence of finger clubbing or platythorax), frequency of complications (e.g. pneumothorax, pneumomediastinum or pulmonary hypertenstion) or causes of death in the study population.
It was my pleassure to read the manuscript. I wish you good luck with publication.
Reviewer 2 Report
General Comments
This is a well written and interesting paper. However revisions are needed to improve the clarity of this work for readers
Introduction:
- The introduction very much focuses on secondary PPFE. As the paper is a comparison piece of secondary and idiopathic PPFE, greater introduction to iPPFE (e.g. prevalence, diagnosis etc.) is needed
- One could argue that PPFE with co-existent lower lobe UIP or ILD is secondary PPFE. Can the authors provide more clarity in the introduction as to why this is not the case
Methods:
- A flow diagram of numbers of cases included and on what basis e.g. histopathology-confirmed vs. imaging only vs. both would be helpful
- The authors have not mentioned in the methods nor specified the rationale for excluding patients who received lung transplantation or had co-existing aspergillus. This is only highlighted in the limitations. Please explain why these patients were excluded and the proportion of idiopathic and secondary PPFE patients excluded for this reason.
- Exclusion of transplanted patients may have biased outcome results as those not transplanted may represent a sicker cohort. Did the authors consider a combined outcome of transplant or death?
- Survival analysis was used with time to death or last contact as the outcome. This suggests an assumption was made that patients potentially lost to follow-up had died. Please comment on the validity and justification for using this approach
Results
- Were all non-UIP ILD cases NSIP or were other patterns detected. Please give breakdown.
- What was the aetiology of non-UIP ILD pattern in idiopathic PPFE patients?
- Please specify the mortality rate overall and for each PPFE group (idiopathic and secondary)
- Given the small numbers consider pooling multivariate analysis to include the entire cohort with adjustment for PPFE type
Table 2: Inclusion of proportions on the table instead of or in addition to absolute numbers would improve visual comparison of group characteristics
Table 3:
- Please show HR (95% CI) and p values for all variables included in analysis even if result non-significant
- Please clarify whether the variable "absence of ILD in the lower lobes" includes patients with non-UIP pattern ILD
Figure 3A: Why was absence of ILD and non-UIP pattern ILD pooled? A comparison of outcomes between PPFE vs PPFE with non-UIP ILD vs PPFE with ILD would be of interest
Discussion:
- Can the authors hypothesise as to why the prognosis of secondary PPFE with UIP may be better
- Comment on why DLCO but not presence of ILD was an independent predictor increased mortality in those with secondary PPFE. What was the suspected aetiology of reduced DLCO - was it related to PPFE itself or a feature of their other disease process e.g. Pulmonary hypertension
- The numbers of patients with secondary PPFE is so small that strong conclusions on mortality risk can not be drawn. This needs to be more clearly acknowledged in the discussion and limitations
